# Responding to Global Learning Needs during a Pandemic: An Analysis of the Trends in Platform Use and Incidence of COVID-19

**Heini Utunen** *, **Richelle George**, **Ngouille Ndiaye**, **Melissa Attias**, **Corentin Piroux** and **Gaya Gamhewage**

World Health Organization, 1211 Geneva, Switzerland; georger@who.int (R.G.); ndiayen@who.int (N.N.); attiasm@who.int (M.A.); corentin.piroux@ieseg.fr (C.P.); gamhewageg@who.int (G.G.)
* Correspondence: utunenh@who.int

**Abstract:** On 11 March 2020, the World Health Organization (WHO) declared the outbreak of coronavirus disease (COVID-19) to be a pandemic. As a result, the OpenWHO.org online platform, which serves as WHO's learning hub for emergencies, was tested for the first time on its core purpose of scaling up trusted public health information in a global emergency. This descriptive study examines and documents the WHO learning response in the early months of the pandemic by comparing epidemiological information and OpenWHO.org use in the countries with the highest COVID-19 cases. Statistical datasets from OpenWHO.org and WHO's COVID-19 dashboard were overlaid for the period 11 March–22 May 2020. During this period, for most of the 24 countries with the highest COVID-19 cases, platform use showed a corresponding trend. Courses published in the official languages spoken in these countries were well utilized, indicating a need to produce materials in languages spoken by affected communities. Of the countries with the highest number of users on OpenWHO, only half were top users of the platform before the pandemic. The existence of an established online platform for health emergencies assisted WHO in massively and quickly scaling up the dissemination of essential learning materials for COVID-19.

**Keywords:** COVID-19; pandemic; online learning; OpenWHO; WHO

## 1. Introduction

The COVID-19 pandemic has affected nearly every country on the planet. The demand for trustworthy information has surged as the virus has spread. As restrictions on travel and physical distancing measures were put into place across the globe, those affected began searching online for information about coronavirus disease [1]. OpenWHO, the WHO Health Emergencies Programme's learning platform, which previously focused on infectious disease work mainly in the Global South, expanded in use throughout the world. In just 10 weeks after WHO declared a Public Health Emergency of International Concern (PHEIC) on 30 January 2020, the platform witnessed course enrolments jump from 177,209 to 1,596,892, an increase of 801.14% [2].

OpenWHO was created in 2017 with emergencies like COVID-19 in mind. During a pandemic, WHO needs to reach millions of people across the globe with real-time learning materials. Courses need to be accessed easily in low bandwidth settings or even where there is no internet [3]. Knowledge must be presented in easy-to-use formats [4]. The self-paced format affords interested learners the flexibility to take courses at their own pace [5].

Research suggests that open online learning, including the fast delivery of knowledge in the context of disease outbreaks, can contribute to public health capacity building [6–9]. Further, a 2018

study from Mexico indicates that people have an increased interest in learning when confronted with an emergency [9]. A pandemic is usually caused by a novel pathogen, which means that much of the knowledge needed to respond did not exist previously. For COVID-19, WHO experts fast-tracked the development of learning material to support the response and launched the first course for COVID-19 on OpenWHO on 26 January 2020, four days before the PHEIC declaration.

While some studies on learning during COVID-19 have emerged, there are no systematic or meta-analyses, including research protocols, focusing on online learning and COVID-19 [10–15].

Some studies refer to mass communication for specific groups, such as youth, through social networks as an efficient method of health-related information dissemination during periods such as lockdowns [16]. Research also suggests that "the Internet is used for adult education in most professional domains, but its use for continuing medical education is less developed" [17], and that Massive Open Online Courses (MOOCs) foster the dissemination of educational content and research results at a large scale and also enable collaborations [18].

MOOCs generally refer to online classes or lectures that offer unlimited registration for anyone who wants to participate, with the open nature differentiating MOOCs from online courses offered for academic credit [7–9]. Although they have been found to have low retention rates and appear to be more popular among participants in North America and Europe, MOOCs also have the potential to make quality educational resources available across geographical and social boundaries, with particular benefit to students in developing countries who otherwise may not have access to such resources [19]. Healthcare MOOCs are useful for a variety of populations, from the general public to specialized and highly experienced professionals, especially during social isolation [20].

For OpenWHO, an immediate priority was making resources available in multiple languages, as research suggests that people prefer online materials in their own language whenever possible, including when searching for health information [21,22]. Further, a 2017 study of MOOCs about disease outbreaks found that translation into more languages could improve participation and traction among affected communities [8]. In order to support the emergency response, a total of 67 courses in 22 languages were quickly delivered during the two months following the pandemic declaration on 11 March 2020.

This study focuses on the use of OpenWHO.org. It examines its use in the countries with the highest number of COVID-19 cases in the 10 weeks following the pandemic declaration, as well as the changes in patterns of use before and after the COVID-19 outbreak.

## 2. Materials and Methods

This descriptive study documents some of the trends characterizing the global audience reached by OpenWHO's open online COVID-19 courses. The aim of this study was to gain a better understanding of the audience that sought access to the WHO learning materials during the coronavirus pandemic. The users' metrics, including location, gender, language, age, and affiliation, are explored to inform the use case of these open online courses and identify the variations in use driven by the pandemic, thus, documenting the platform's first-ever public health digital response in the face of a global pandemic. This study also investigates the use of the OpenWHO COVID-19 digitized learning materials in the areas most affected by the virus to assess the efficacy of the geographic reach of these open online resources.

Anonymized statistical datasets were obtained from the OpenWHO integrated reporting system, providing platform-wide, topical, and course-specific (language version) datasets. As of 22 May 2020, OpenWHO offered 10 COVID-19 response-related topics, with most available in several languages: (1) Emerging respiratory viruses, including COVID-19: methods for detection, prevention, response, and control; (2) ePROTECT Respiratory Infections; (3) Infection Prevention and Control (IPC) for COVID-19 Virus; (4) WHO Clinical Care Severe Acute Respiratory Infection Training; (5) COVID-19: How to put on and remove personal protective equipment (PPE); (6) Standard precautions: Hand hygiene; (7) Severe Acute Respiratory Infection (SARI) Treatment Facility Design; (8) COVID-19:

Operational Planning Guidelines and COVID-19 Partners Platform to support country preparedness and response; (9) Standard precautions: Waste management; and (10) Introduction to Go.Data-Field data collection, chains of transmission and contact follow-up.

Additional linguistic analysis was conducted on the most popular and translated resource, the introductory course: Emerging respiratory viruses, including COVID-19: methods for detection, prevention, response and control; this was OpenWHO's first COVID-19 course, which was published on 26 January 2020, and totaled 459,412 enrolments as of 22 May 2020. Analysis was conducted to assess enrolment proportions in the 22 languages in which it was available and to examine the top language versions used by countries most affected by the pandemic. This informs the relevance of the multilingual aspect of a public health knowledge transfer endeavor.

Course users' OpenWHO activity by geographic location was captured by Google Analytics. A user's activity was measured by web session; with an active session defined as the period of time a user is actively engaged with the OpenWHO platform (via desktop, mobile, or app). The statistical dataset obtained on OpenWHO usage was overlaid with a statistical dataset from the WHO Health Emergencies Programme COVID-19 dashboard, including the countries with the highest number of COVID-19 cases from 11 March–22 May 2020 [23].

Microsoft Excel was used to conduct the user analysis to provide descriptive statistics assessing user metrics and patterns. Frequencies were calculated for the following variables: affiliation, gender, language, age, and geographic location by country and aggregated by WHO regions. This information is disclosed by the learner during the registration process for the OpenWHO platform. The patterns identified help characterize the trends within this global audience. Further analysis of exceptional online user groups, such as 70+ year-old users, was performed to investigate their specific use of the COVID-19 online resources.

## 3. Results

### 3.1. Countries with the Highest Number of COVID-19 Cases and Use of the OpenWHO Platform

As the COVID-19 epidemic grew, the need for learning materials surged accordingly. The countries that experienced the highest numbers of confirmed COVID-19 cases during the 10-week period following the pandemic declaration (11 March–22 May 2020) also had a similar number of users on OpenWHO (Figure 1). The only major exceptions were the USA and India where the two figures diverged more significantly. In the USA, the case count was the highest globally, and use of OpenWHO was also high in comparison to other countries. India has historically exhibited the greatest use of OpenWHO both before and after the pandemic declaration. With an already high number of users before the pandemic, OpenWHO use in India remained elevated during the pandemic, making up 15% of the total use of the platform.

While most of the countries in this list experienced similar levels of COVID-19 cases and OpenWHO platform use, Russia and China exhibited the lowest platform use despite their COVID-19 cases. Users from these countries have not historically been active users on OpenWHO, nor have there been materials in their languages on the platform before the pandemic, with the exception of one course in Russian. There is also a barrier to accessing the platform in China, as the platform's webpages are not accessible to China-based users. The number of users in Brazil, the UK, Italy, and Spain is similar to the COVID-19 case count.

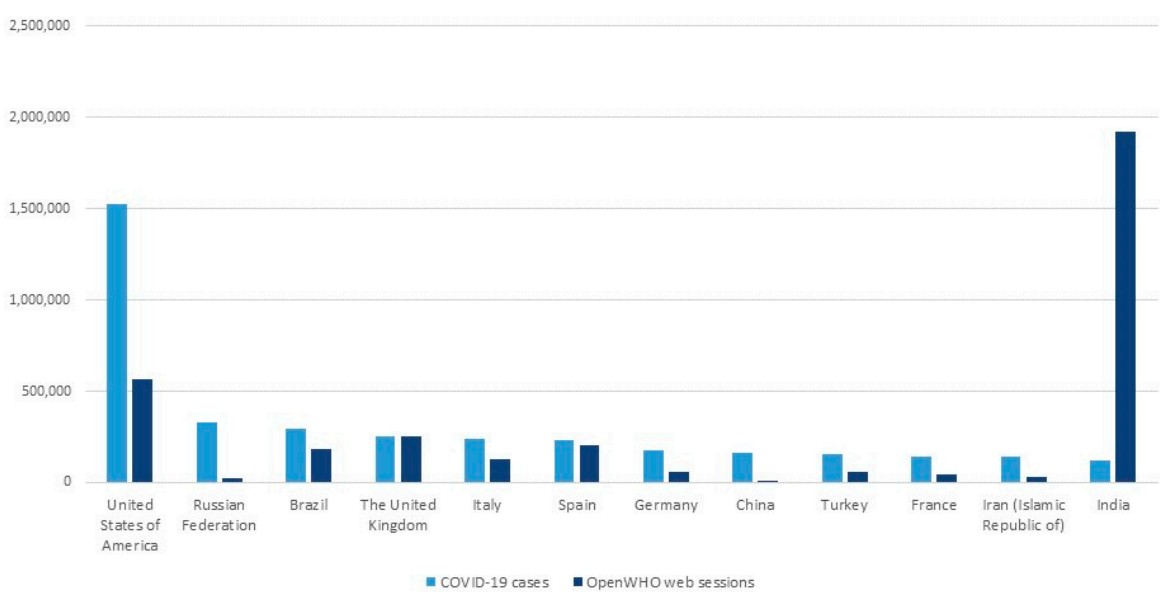

**Figure 1.** OpenWHO web sessions in the 12 countries with the highest cumulative confirmed COVID-19 cases (11 March–22 May 2020).

The countries with the 13th to 24th highest cumulative confirmed COVID-19 cases (Figure 2) largely differ in trend to those in Figure 1. OpenWHO use in Peru, Canada, Saudi Arabia, Mexico, Chile, Pakistan, and Ecuador is high, with all countries featuring among the top 24 countries with regard to OpenWHO use after the pandemic declaration. Belgium, The Netherlands, Qatar, and Sweden, on the other hand, show a similar trend to most countries illustrated in Figure 1.

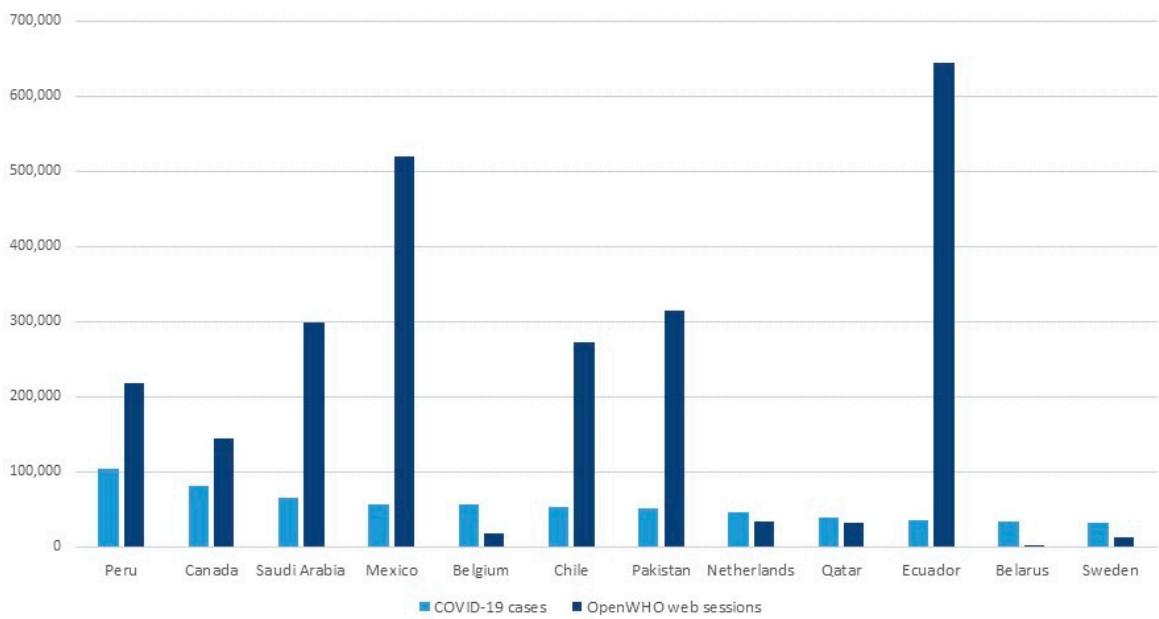

**Figure 2.** Web sessions in the countries listed in positions 13–24 in the highest cumulative confirmed COVID-19 cases list (11 March–22 May 2020).

## 3.2. Regional Use

The same data sets displayed according to WHO region also illustrate interesting geographic trends (Figure 3).

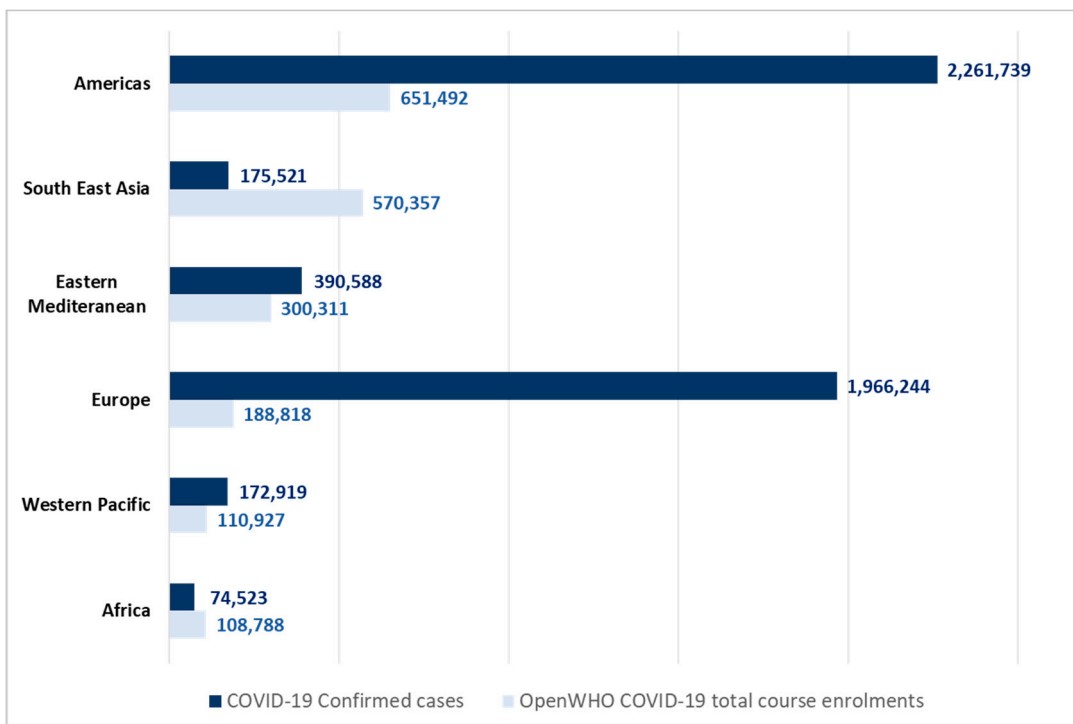

**Figure 3.** Confirmed COVID-19 cases and OpenWHO use (as measured by course enrolments) per WHO region as of 22 May 2020.

As per the confirmed cases count, the WHO Region of the Americas (AMRO) was the epicenter of the disease during the research period and also has the highest number of enrolments in OpenWHO courses. The COVID-19 case count in the European Region (EURO) is the second highest globally, though enrolments from this region are relatively low, reaching fourth place among the WHO regions. The South-East Asian (SEARO) and Eastern Mediterranean (EMRO) regions have a higher number of enrolments than the European region, although their case count is lower. The number of enrolments from SEARO was fueled by exponentially high use from India. Overall, for the African (AFRO), Eastern Mediterranean, South-East Asian, and Western Pacific (WPRO) regions, which are for the most part concentrated in the Global South, the levels of OpenWHO use and confirmed COVID-19 cases are close.

### 3.3. Course Use by the Languages of Affected Populations

WHO prioritized publishing learning materials about COVID-19 in the languages spoken in the countries most affected by the spread of the disease [24,25]. By 22 May 2020, the introductory COVID-19 course was published in 22 languages and hosted 459,412 enrolments. The first 12 courses published account for 98.88% (n = 454,289/459,412) of the total course enrolments for all 22 languages combined (Figure 4).

An examination of the use of each language version of the introductory course reveals that users from the countries most affected by the pandemic accessed this course in the national language of their country, as illustrated in Table 1. Collectively, the three most commonly used language versions in each country account for more than 90% of the total course use from that country. This is the case for all 12 countries with the highest number of confirmed COVID-19 cases, with the exception of Russia, where a larger variety of language versions of the course were used. The Indian Sign Language course was among the top language versions used in the USA and China, suggesting that there are populations in each country in need of this resource. Less surprisingly, the course also features in the top three in India. In total, this course hosted 12,561 enrolments, making it the fourth most popular language version of the introductory course.

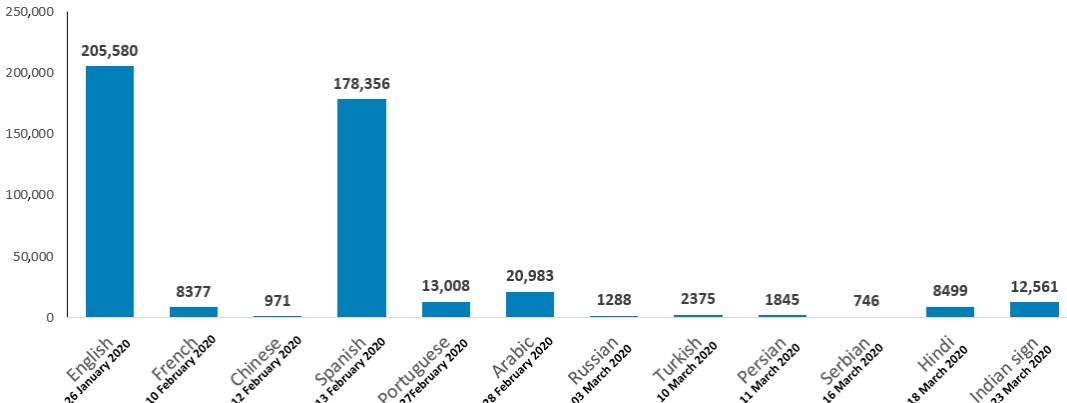

**Figure 4.** First 12 language versions of the introductory COVID-19 course available and enrolment numbers.

**Table 1.** Language versions of the introductory course used in the 12 countries with highest confirmed COVID-19 cases. Top three language versions used listed per country.

| Country | Top Language Version Used, n (%) | Second Most Used Language Version, n (%) | Third Most Used Language Version, n (%) | Total Top Three Language Versions Used, n (%) |
|---|---|---|---|---|
| **USA** | English 22,693/25,070 (90.52%) | Spanish 1324 (5.28%) | Indian Sign Language 216 (0.86%) | 24233 (96.66%) |
| **Russia** | English 246/584 (42.12%) | Russian 160 (27.40%) | Arabic 98 (16.78%) | 504 (86.30%) |
| **Brazil** | Portuguese 4642/6178 (75.14%) | English 1149 (18.60%) | Spanish 316 (5.11%) | 6107 (98.85%) |
| **UK** | English 6561/7125 (92.08%) | Arabic 119 (1.67%) | Spanish 98 (1.38%) | 6778 (95.13%) |
| **Italy** | English 1214/1472 (82.47%) | Spanish 95 (6.45%) | French 59 (4.01%) | 1368 (92.93%) |
| **Spain** | Spanish 6039/7870 (76.73%) | English 1680 (21.35%) | Portuguese 62 (0.79%) | 7781 (98.87%) |
| **Germany** | English 1092/1560 (70.00%) | Spanish 191 (12.24%) | Arabic 124 (7.95%) | 1407 (90.19%) |
| **China** | English 214/342 (62.47%) | Chinese 108 (31.58%) | Indian Sign Language 7 (2.05%) | 329 (96.20%) |
| **Turkey** | Arabic 917/2275 (40.31%) | English 660 (29.01%) | Turkish 617 (27.12%) | 2194 (96.44%) |
| **France** | French 892/1532 (58.22%) | English 475 (31.01%) | Arabic 51 (3.33%) | 1418 (92.56%) |
| **Iran (Islamic Republic of)** | Persian 288/574 (46.69%) | English 224 (39.02%) | Arabic 28 (4.88%) | 520 (90.59%) |
| **India** | English 38,043/52,577 (72.36%) | Hindi 6663 (12.67%) | Indian Sign Language/5970 (11.35%) | 50,676 (96.38%) |

In Brazil, France, India, Iran, Spain, the UK, and the USA, the most used language version of the introductory course was the language that is most widely spoken or an official national language in that country. English was also the most popular introductory course language in India, Germany, and Russia. The Hindi course is the second most popular course language in India. The introductory course in German was published after 22 May 2020.

Though use of the platform is in general more modest in China and Russia, Chinese and Russian are the second most popular language version in each country respectively. Arabic is among the top three language choices in France, Germany, Turkey, Russia, and the UK, pointing to a globalized audience who, regardless of where they reside, prefer to learn in their native languages.

Of the languages mentioned above, Indian Sign Language, Turkish, Hindi, and Persian were added to the platform for the first time when published as language versions of the introductory course. With regard to the Persian course, 49.84% (n = 920/1845) of enrolments were from users who were new to the platform. For the Hindi course, this number was 29.58% (n = 2514/8499), compared to 57.17% (n = 1358/2375) for the Turkish course and 15.53% (n = 1951/12,561) for the Indian Sign Language course. Of all the language versions of the introductory course, the Spanish (65.18%, n = 116,250/178,356), Turkish (57.17%, n = 1358/2375), Portuguese (51.96%, n = 6759/13,008), Persian (49.84%, n = 920/1845), and Arabic (49.52%, n = 10,390/20,983) courses had the highest proportion of new users. The average proportion of new users across all language versions of the introductory course was 52.64% (n = 241,840/459,412).

### 3.4. 70+ User Group

Following the outbreak of COVID-19, the OpenWHO platform experienced a significant increase in use from individuals aged 70+. When comparing the user age distribution before the pandemic and immediately after the pandemic declaration, one of the age groups that witnessed the largest change is the 70+ group. Before the outbreak, the 70+ user group consisted of 116 users accounting for only about 0.0025% (n = 112/44,449) of the platform-wide total of 86,000 users in December. By May 2020, the proportion of the 70+ age group had increased to 5.14% of the total registered users, representing about more than 76,000 of the total 1,484,163 users registered on the platform (Figure 5). Currently, the proportion of learners aged 70+ is four times greater than the 60–69 age group (1.29%) and is comparable to the number of learners in the under 20 (5.38%) and 50–59 (5.06%) age groups.

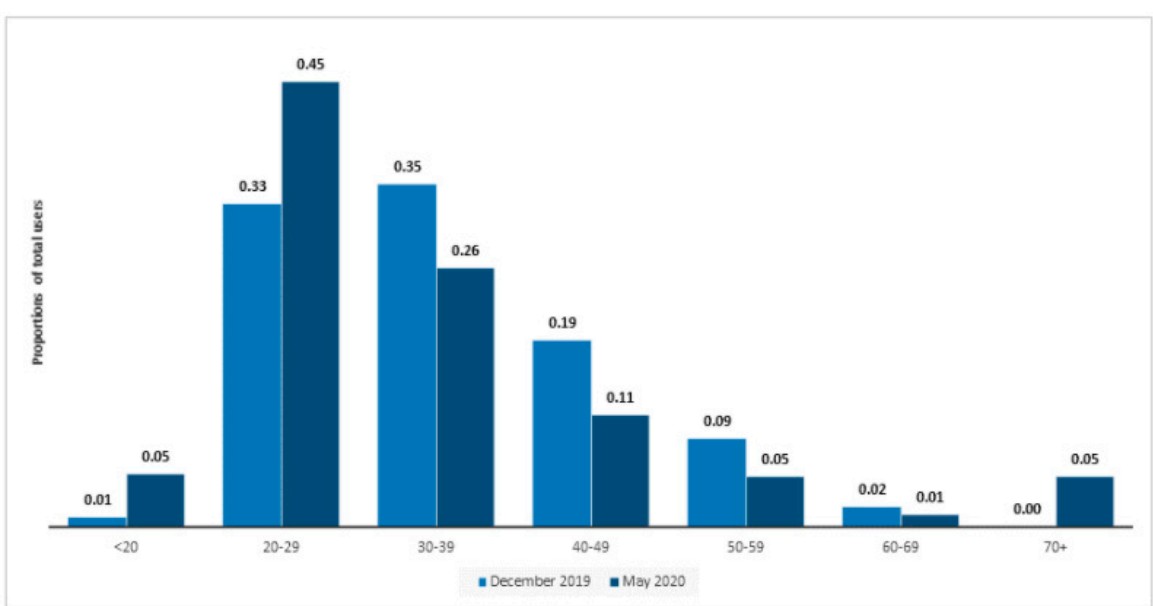

**Figure 5.** OpenWHO user age distribution before and post COVID-19 pandemic.

A natural explanation for this change would be the impact of COVID-19 on elderly people. Dr Hans Henri P. Kluge, WHO Regional Director for Europe, stated that, "Some of the reasons older people are greatly impacted by COVID-19 include the physiological changes associated with ageing, decreased immune function and multimorbidity which expose older adults to be more susceptible to the infection itself and make them more likely to suffer severely from COVID-19 disease and more serious complications" [26]. Research suggests that online learning can be a useful tool for informing elderly individuals familiar with computers and that solving problems faced in their lives is a popular motivation for older adults in MOOCs [27,28]. Research also discusses the specific design needs for providing barrier-free and optimization of online learning for the elderly and also focuses on active learning aspects among elderly people [29,30].

Out of the 10 courses that address the preparation and response to COVID-19 on OpenWHO, the introductory COVID-19 course, available in 22 languages and providing basic information about the novel coronavirus, exhibits the highest proportion of learners within the age group of 70+ (10.30%, n = 23,852/231,475), which is more than double the average enrolment from this age group on OpenWHO in general (5.14%). Interestingly, the other most generic course, on prevention and control, also witnessed great popularity among these older adult learners (9.21%, n = 26,644/289,371), while more technical courses, primarily aimed at frontline responders, have significantly lower than average proportions (Figure 6).

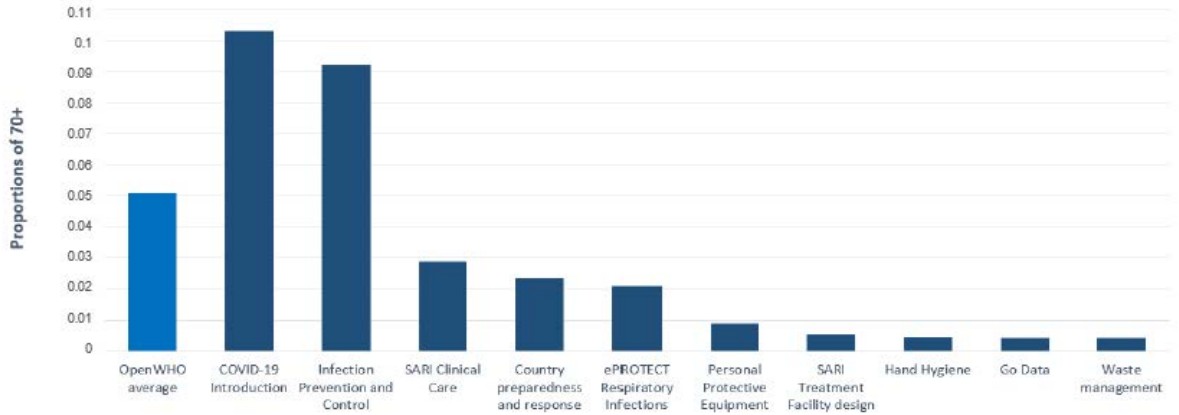

**Figure 6.** OpenWHO COVID-19 online course use among the 70+ age group.

For the Spanish version of the introductory course, one out of five course users is over 70 (Figure 7). In fact, the Spanish version of the course accounts for 80.99% (n = 19,317/23,852) of the total enrolments from users aged 70+ across all language versions of the introductory course, followed by the English version (13.25%, (n = 3160/23,852). Geographic analysis of the 70+ learner group (who are predominantly Spanish speaking and a majority of whom are female (62.70%, n = 11,352/18,105), reveals that 97.32% (n = 20,027/20,579) accessed the course from the WHO Region of the Americas, particularly from Ecuador (46.50%, n = 9570/20,579), Colombia (10.86%, n = 2234/20,579), Argentina (9.71%, n = 1999/20,579), Mexico (9.62%, n = 1980/20,579), and Chile (9.33%, n = 1920/20,579).

### 3.5. Change in Geographic Platform Use Patterns over the Course of the Pandemic

Thirteen out of the 24 countries with the highest number of COVID-19 cases between 11 March and 22 May 2020 were also in the top 24 countries in terms of OpenWHO use in the same period. Additionally, out of the 24 countries with the greatest OpenWHO use prior to the pandemic, 10 of these would go on to join the list of the 24 countries with the highest number of COVID-19 cases during the pandemic (Table 2).

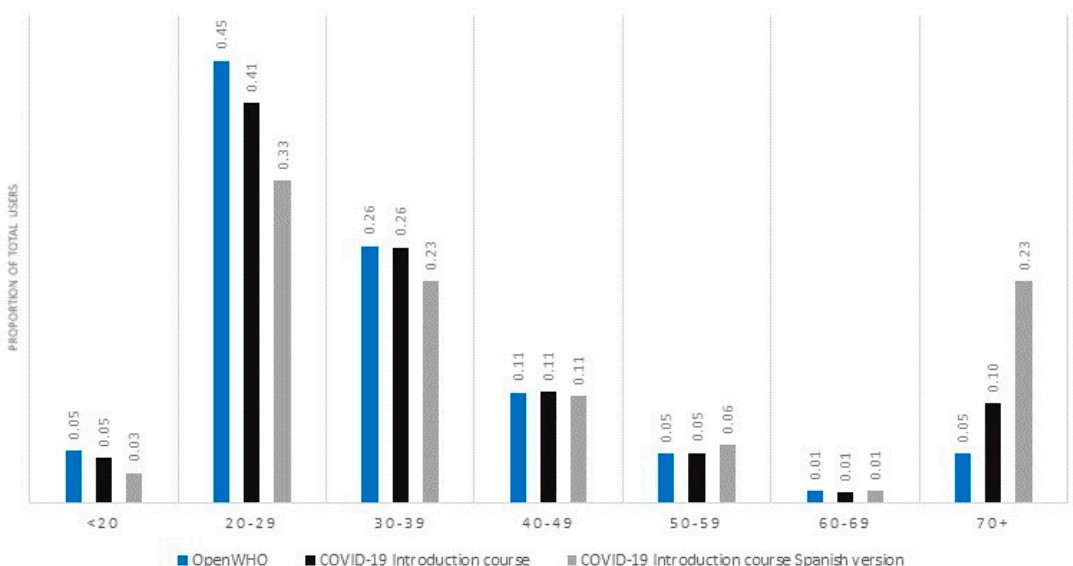

**Figure 7.** Age distribution of users enrolled in the COVID-19 introductory course on OpenWHO.

**Table 2.** Top 24 countries by OpenWHO use during the last 10 weeks of 2019 and the 10 weeks following the COVID-19 pandemic declaration.

| Final 10 Weeks of 2019 (23 October–31 December 2019) | 10 Weeks Following the Pandemic Declaration (11 March–22 May 2020) |
|---|---|
| 1. India | 1. India (0) * |
| 2. Nigeria | 2. Ecuador (-) * |
| 3. USA | 3. USA (0) * |
| 4. Democratic Republic of the Congo | 4. Mexico (-) * |
| 5. UK | 5. Bangladesh (-) |
| 6. Portugal | 6. Colombia (-) |
| 7. Egypt | 7. Argentina (- |
| 8. Canada | 8. Pakistan (+6) * |
| 9. Saudi Arabia | 9. Saudi Arabia (0) * |
| 10. Sudan | 10. Nigeria (−8) |
| 11. Kenya | 11. Chile (-) * |
| 12. Switzerland | 12. Philippines (+5) |
| 13. South Sudan | 13. Iraq (-) |
| 14. Pakistan | 14. UK (−9) * |
| 15. France | 15. Peru (-) * |
| 16. Germany | 16. Egypt (−9) |
| 17. Philippines | 17. Spain (-) * |
| 18. Ethiopia | 18. Brazil (+3) * |
| 19. The Netherlands | 19. Canada (−11) * |
| 20. Cameroon | 20. Nepal (-) |
| 21. Brazil | 21. Indonesia (-) |
| 22. Australia | 22. Italy (-) * |
| 23. Uganda | 23. Australia (−1) |
| 24. Côte d'Ivoire | 24. United Arab Emirates (-) |

* Countries in the top 24 for confirmed COVID-19 cases; (-) New to the top countries list for OpenWHO use after the pandemic declaration.

It is also interesting to note the change in the geographical pattern of platform use before the pandemic and in the 10 weeks following the pandemic declaration. Half of the top 24 countries with the greatest platform use between 11 March–22 May 2020 were already established as such prior to the pandemic, while the other half joined the list during the pandemic. Three major findings can be drawn from examining this data:

(1)    Before the pandemic, 10 African countries were among the top 24 countries in terms of OpenWHO use. In the 10 weeks after the pandemic was declared, the use among African countries fell and, as a result, only two countries, Nigeria and Egypt, remained in the top 24 list. In the same period, no African countries were present among the 24 countries with the highest number of confirmed COVID-19 cases.

(2)    Six South American countries and three Asian countries joined the top 24 list for the first time. Four of these six South American countries were among the countries with the 13th to 24th highest number of confirmed cases during this period.

(3)    Before the pandemic was declared, six European countries appeared in the top 24 countries list. Following the pandemic declaration, only Italy, Spain, and the UK appeared in the list. Of these, Spain and Italy appeared as new additions to the list. Each of these three countries was among the top countries globally in terms of confirmed cases, situated in positions 4–6.

## 4. Discussion

The aim of this study was to investigate the properties of users accessing OpenWHO's COVID-19 courses during the pandemic. The findings illustrate that the platform is used most in the countries with the highest transmission of the virus. There was a 54.17% (n = 13/24) overlap between the list of the 24 countries with the highest confirmed COVID-19 cases between 11 March–22 May 2020 and the 24 countries with the highest use of the OpenWHO platform in the same period.

Though the use of the platform is mixed among the WHO regions, a comparison of the COVID-19 case count in each region against the level of OpenWHO use revealed some interesting trends. Most notably, the use of OpenWHO was highest in the Region of the Americas, where the number of confirmed COVID-19 cases was also the highest. This fact, in addition to the close strong relationship between the number of confirmed COVID-19 cases and OpenWHO use in the African, Eastern Mediterranean, South-East Asian, and Western Pacific regions, supports the argument that the learning interventions offered on OpenWHO are of use to populations affected by the outbreak. The location of the latter four regions in the Global South also confirms the platform's usefulness and reach in resource-limited contexts.

In the majority of the countries most affected by the pandemic, OpenWHO users accessed materials in the languages most widely spoken in their locations. On average 52.64% (n = 241,840/459,412) of users who enrolled in the introductory course (across all language versions) were new to the platform, suggesting that a significant proportion of users arrived on the platform for the purpose of gaining essential knowledge about the pandemic. The proportion of new users enrolled in the Spanish and Turkish introductory courses was above average, highlighting the importance of providing materials in these languages to access new audiences. The high uptake of courses in countries in the Region of the Americas could be further explained by the early translation of the introductory course into Spanish as well as support in outreach through collaboration with the Pan American Health Organization (PAHO) Virtual Campus for Public Health. A similar proportion of new users can be seen in several other language versions, including Persian, Serbian, and Macedonian, which, like Turkish, are languages new to the platform.

The COVID-19 pandemic increased the proportion of users aged 70+ on the platform. The introductory course exhibited the highest proportion of 70+ users of any COVID-19 course, exceeding the platform average. The increase in platform use among users aged 70+ could be explained by the heightened impact of the disease on older adults, who are deemed high-risk.

With regard to the geographic change in OpenWHO use during the pandemic, 11 out of the 13 countries that were new entrants into the top 24 list are in the Global South. Of these, nine are low- or middle-income countries according to the World Bank. In addition, as the platform use expanded into the Spanish-speaking world, six South American countries entered the top 24 list. This reflects a significant shift in the pattern of platform use from 2019. This finding was also reflected in an analysis of data extracted from the two months after the launch of the introductory course, which found that apart from the English course, the Spanish version was overwhelmingly the most popular [31].

Prior to the pandemic, the high levels of OpenWHO use in African countries reflected the concentration of health emergencies within the continent. The change in the pattern of OpenWHO use has mirrored the way that the spread of COVID-19 has shifted the loci of outbreaks to other geographical regions. The two African countries that remain in the list of the top 24 countries for OpenWHO use are also among the three countries with the largest populations on the African continent. The countries in the top 24 list constitute 55.74% (n = 5,993,930/10,752,536) of the total use of OpenWHO during the research period.

In summary, those who require learning support most, whether due to their location at an epicenter of the pandemic or due to their age, have accessed the platform. Further research will include longitudinal comparisons of the presence of COVID-19 and the use of OpenWHO in select countries.

Limitations: The focus of this article is the first 2.5 months following the pandemic declaration. Particular attention was given to the countries with the highest confirmed COVID-19 cases. Users in some countries are unable to access the platform and, thus, their use is difficult to compare. Some materials have also been embargoed through different channels and, thus, not all countries and courses are fully comparable. It should also be noted that course enrollees' proportions are determined based on the users who disclose the relevant information during the OpenWHO registration process.

## 5. Conclusions

This study suggests a real-time strong relationship between the significant increase in user numbers on an established WHO online learning platform and the number of cases of COVID-19 in the 24 countries with the highest number of COVID-19 cases during the study period. It also provides evidence on the value of offering learning material in the native languages of the people at the center of emergencies. Additionally, it illustrates that even when courses are primarily intended for responders, the segment of the public most at risk, for example, the elderly in the case of COVID-19, will also access the learning content.

The evidence illustrates the need to establish formal processes and increase investment with regard to the practice of fast-tracking the development and delivery of courses during an emergency; the need to focus on accessibility issues, such as translation into the languages spoken in the epidemic or pandemic hotspots; and the need to offer learning material tailored to the people at risk, in addition to responders. While further research into all three aspects is needed, the research presented here is unique in that it analyses data collected during a pandemic and links epidemiological trends with the uptake of learning online. On a broader level, it underscores the importance of real-time training during a disease outbreak, epidemic or pandemic, with heightened focus on accessibility and the preferences of users.

**Author Contributions:** Study design: H.U.; Data collection, figures, data analysis: N.N., C.P., R.G.; Data interpretation, writing: H.U., G.G., R.G., M.A., N.N.; Literature search: M.A. All authors have read and agreed to the published version of the manuscript.

**Funding:** This research received no external funding.

**Acknowledgments:** The OpenWHO platform operations have been made possible by the strong efforts of Oliver Stucke. The authors would also like to express their gratitude to Susan Spackman for the editorial support and Anna Tokar for the literature searches. The entire OpenWHO team from the Learning and Capacity Development Unit for making the pandemic learning response online possible.

**Conflicts of Interest:** The authors declare no conflict of interest.

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
