# Peer review of "Responding to Global Learning Needs during a Pandemic: An Analysis of the Trends in Platform Use and Incidence of COVID-19"

_education, doi:10.3390/educsci10110345_

Round 1

Reviewer 1 Report

General recommendations

The findings of this study may provide useful information for reorienting practices of massive open online courses on health education issues in a situation of pandemic. Nevertheless, major revision is required.

Abstract

The abstract should contain a background before the aim of the study. It should also specify the type of design used in the study.

Introduction Section

The introduction should explain what MOOCs are, their advantages and disadvantages, and what are their differential characteristics with respect to other online educational methodologies. 

Explain the acronym the first time they are named (p.e. MOOC see lines 86 and 92-93).

Methods Section

The methodology section is too brief. It should contain several sections, pe:

  • Research context.
  • Research design.
  • Elements of analysis and their description.

In this section it is said where the pandemic data are obtained from, but does not specify the URL of the data on students enrolled in the various courses. This is interesting to facilitate the replication of the study.

The research design used in the study should be made explicit.

In Statistics section, the specific software used for data analysis should be specified. In addition, it would be interesting to carry out studies of the degree of association between the variables analyzed and not only explain basic statistics referring to percentages.

Results, Discussion and Conclusion Sections

The results are interesting, but it is missing that statistics are presented that allow establishing levels of association between variables, or hypothesis contrast tests that allow establishing quantitative data on the magnitude of the difference between certain variables. Note that expressions of the type are used:

Lines 21 y 22: The only major exceptions are the USA and India where the two figures diverged more significantly.

Lines 276 – 279: This fact, in addition to the close correlation between the number of confirmed COVID-19 cases and OpenWHO use in the African, Eastern Mediterranean, South-East Asian and Western Pacific regions, AFRO, EMRO, SEARO and WPRO regions, supports the argument that the learning interventions offered on OpenWHO are of use to populations affected by the outbreak…

Lines 285 y 286: suggesting that a significant proportion of users arrived at the platform for the purpose of gaining essential knowledge about the pandemic.

Lines 291-293: A similar proportion of new users can be seen in several other language versions, including Persian, Serbian and Macedonian, which, like Turkish, are languages new to the platform.

Lines 301 y 302: This reflects a significant shift in the pattern of platform use from 2019.

Finally, the discussion should begin by referring to the goal of the study.

Reviewer 2 Report

The paper deals with the analysis of the intensity of use of the online platform OpenWHO.org for learning topics related to COVID 19 in the countries with the most COVID-19 cases.  

The manuscript is aligned with the aims and scope of the journal, but it is not scientifically grounded enough, the applied methods and the data analysis are elementary. Although the topic is current, the paper focuses only on statistical analysis of frequencies and is more reminiscent of a statistical report than a scientific paper. This is the main and essential objection.

The implications of the research, as well as its significance, are not clearly stated.

In addition, the data were analyzed in absolute terms and not relative to the population of the countries covered by the analysis.

Although an increase in the platform utilization rate by two categories (20-29 years old and 70+) was identified, only the segment 70+ was analyzed.

Some technical remarks:

  • As is noted in instructions for authors, “The abstract should be a single paragraph and should follow the style of structured abstracts, but without headings”. Thus, authors should remove some words from abstract section (such as background, methods, results...)
  • style of referencing is not in accordance with the instructions.
  • Figure 4 has two different captions: “Figure 4. Use of the first 12 language versions of the introductory course, organized by the date of 157 course launch and illustrating the number of enrolments. Figure 4 First 12 language versions of the 158 introductory COVID-18 course and the enrolment numbers”
  • In many places the percentages are poorly written, for example (49 · 84%) instead of (49.84%)
  • The second sentence in the title of Figure 5 should be moved to the main text. “Figure 5. OpenWHO user age distribution before and post COVID-19 pandemic. Major change in the OpenWHO user age distribution following the pandemic was witnessed in the groups less than 30 199 years and more than 70 years of age.”

Author Response

Many thanks for the various comments and contributions from the reviewers. We have addressed all the aspects of the reviewers comments except for not going into the suggested mathematical modelling as this study had been decided to be descriptive by the nature. The existing study results speak to the article focus. The research method is not based on inferential statistics but on descriptive statistics meant to provide metrics and patterns to characterize the OpenWHO course user database. The objective of this descriptive study is to better characterize the global audience of the OpenWHO COVID-19 resources. Indeed, the identified correlations can be explored and tested in another study probing into these apparent correlations.

The analysis conducted assesses frequencies found for variables like age, gender, language or user location as well as comparing OpenWHO activity trends to the areas most affected by COVID-19 cases.

All other review comments have been incorporated.

Reviewer 3 Report

COMMENTS AND SUGGESTIONS:

The paper is readable and mostly adapted to the author's instructions (template file). Paper structure is organized through multiple sections (titles), so the form of paper is mostly clear.

The research method expresses the impression of ordinary comparisons, without a strong mathematical model!

Results are presented very clear through the diagrams and tables. These results are very significant and give a very clear picture of the application of the OpenWHO platform in a population that was more or less vulnerable to the SARS-COV-2 virus. It is possible to see the importance of developing such an approach in order to educate and prevent the population. Hence, this research is original research!

The references are correct, but the visual structure is necessary to prepare according to Template file!

 My opinion is that the paper is acceptable with the corrections!

 Comment 1:

According to Template file, suggestion is to put contents of sections: Evidence before this study, Added - value of this study, Implications of all the available evidence, in the Introduction section. See the Template file!

Comment 2:

Try to reduce Abstract. According to Template file, abstract needs to have about 200 words maximum. Please try to reduce number of words in Abstract section!

Comment 3:

In the text, reference numbers should be placed in square brackets [ ]. Try to fix it, especially in the Introduction section!

Comment 4:

Line: 158

Wrong: …COVID-18…

Correct: …COVID-19…

Comment 5:

Line: 162

Wrong: …22 languages combined. (Figure 4).

Correct: …22 languages combined (Figure 4).

Comment 6:

Table 1 is possible partially place to the previous page!

Comment 7:

Try to place the name of Figure 5 on the same page where the figure is placed!

Comment 8:

Line: 208

Wrong: …for only about 0·0025%...

Correct: … for only about 0.0025%...

Comment 9:

Line: 209

Wrong: … 5·14%...

Correct: … 5.14%...

Comment 10:

Line: 212

Wrong: … (5·38%)...(5·06%)…

Correct: …  (5.38%)...(5.06%)…

Round 2

Reviewer 1 Report

Thanks to the authors for following my suggestions, I hope they have served to improve the quality of the manuscript.   I would just like to add a comment. I would recommend being cautious about using expressions that refer to statistical calculations that are not supported by numerical data. For example, expressions like the one in the conclusion:   This study suggests a real-time CORRELATION between the significant increase in user numbers on an established WHO online learning platform and the number of cases of COVID-19 in the 24 countries with the highest number of COVID-19 cases during the study period.   The term CORRELATION must be supported by a numerical value. I think it would be preferable to use a more qualitative expression such as:   This study suggests a real-time CLOSE RELATIONSHIP between the significant increase in user numbers on  an established WHO online learning platform and the number of cases of COVID-19 in the 24 countries with the highest number of COVID-19 cases during the study period.   Finally, I suggest to the authors to carry out a thoroughly review for minor failures and congratulate them for their effort.      

Author Response

Dear Reviewer, many thanks for this detailed and profound review and recommendations to The numerical terminology such as correlation has been replaced by more qualitative terms as per your suggestions. The whole article has been copy edited after the revisions. Best wishes and the gratitude for your advice from us the authors 

Reviewer 2 Report

The authors have made an important effort to improve their document. Both the most important concerns and the minor formal aspects have been resolved.

Author Response

Many thanks for your consideration, profound advice and detail in the comments. We look forward working forward in the improved article submission. Best wishes, authors.